# Association between Meal Frequency and Weight Status in Spanish Children: A Prospective Cohort Study

**DOI:** 10.3390/nu15040870

**Published:** 2023-02-08

**Authors:** Charlotte Juton, Paula Berruezo, Silvia Torres, Olga Castañer, Genís Según, Montserrat Fitó, Clara Homs, Santiago F. Gómez, Helmut Schröder

**Affiliations:** 1Endocrinology Department, Institut de Recerca Sant Joan de Déu, 08950 Barcelona, Spain; 2Gasol Foundation Europe, Sant Boi de Llobregat, 08830 Barcelona, Spain; 3Faculty of Health Science and Wellbeing, University of Vic-University Central of Catalonia, 08500 Barcelona, Spain; 4Cardiovascular Risk and Nutrition Research Group (CARIN), IMIM (Hospital del Mar Medical Research Institute), 08003 Barcelona, Spain; 5University of Lleida, 25003 Lleida, Spain; 6CIBER Physiopathology of Obesity and Nutrition (CIBERobn), Instituto de Salud Carlos III, 28029 Madrid, Spain; 7Global Research on Wellbeing (GRoW) Research Group, Blanquerna School of Health Sciences, University Ramon Llull, 08025 Barcelona, Spain; 8GREpS, Health Education Research Group, Nursing and Physiotherapy Department, University of Lleida, 25008 Lleida, Spain; 9CIBER Epidemiology and Public Health (CIBERESP), Carlos III Health Institute, 28029 Madrid, Spain

**Keywords:** meal frequency, weight outcomes, children

## Abstract

Childhood obesity is a major international problem, and unhealthy eating habits remain widespread. Increasing the frequency of meals of nutritious food can help children to regulate their appetite and maintain a healthy weight. However, there is scarce prospective evidence on the relationship between the meal frequency and weight outcomes. Therefore, the objective of the present study was to determine the prospective association between the meal frequency, body mass index, and waist circumference in Spanish children. Additionally, we analyzed the impact of the meal frequency on the incidence of excessive weight and abdominal obesity. The study included 1400 children with a mean (SD) age of 10.1 (0.6) and an average follow-up of 15 months. Anthropometric measurements, including the body weight, height, and waist circumference, were measured by trained personnel, and children were asked about whether they usually had the following meals: breakfast, a mid-morning snack, lunch, an afternoon snack, and dinner. Multiple linear regression models revealed a significant (*p* < 0.05) inverse association between the meal frequency with a standardized BMI (zBMI) and the waist-to-height ratio (WHtR) after adjusting for sex, age, allocation to an intervention group, school, maternal education, physical activity, diet quality, and for the corresponding outcome variable at the baseline. Furthermore, the odds of developing abdominal obesity or excessive weight during the follow-up significantly decreased with an increase in the meal frequency after controlling for the same confounders. In conclusion, a higher meal frequency at the baseline was predictive for a lower zBMI, WHtR, and odds of the incidence of excessive weight and abdominal obesity.

## 1. Introduction

Childhood obesity remains a major concern worldwide, with 389 million children and adolescents under the age of 19 presenting with excessive weight [1]. Eating nutritious food and adopting healthy, sustainable dietary habits from an early age are key for maintaining a healthy weight. However, the World Health Organization (WHO) reported poor eating habits in children, including low intakes of fruit and vegetables and a high consumption of sweet and salty snacks [2]. The diet quality of parents and children has been shown to be directly associated, suggesting that the eating behaviors of children are likely to stem from parental dietary practices [3]. Restrictive feeding practices in children can increase the desire for restricted food as well as eating in the absence of hunger and, ultimately, a higher calorie intake [4,5,6,7]. Increasing the frequency of healthy meals/snacks can instead be beneficial as it helps to regulate the appetite and ward off hunger frustration [3]. In children under 5 years of age, longitudinal studies have shown no significant association or very little clinical significance between the meal frequency and the weight status [8,9]. When including older participants, a meta-analysis of 10 cross-sectional studies and 1 case-control study indicated that a higher eating frequency improved the anthropometric status in boys aged 2 to 18 years old [10]. The results for girls are more controversial due to a high heterogeneity between studies, but a longitudinal study reported a significant association [10,11]. Considering the scarce evidence on this topic, more observational studies, ideally with a prospective design, are needed to draw clearer conclusions. Therefore, the objective of the present study was to determine the prospective association between the meal frequency, body mass index, and waist circumference and the incidence of excessive weight and abdominal obesity in Spanish children after an average follow-up of 15 months.

## 2. Materials and Methods

### 2.1. Study Design

The current study was a prospective cohort in the framework of the POIBC study (the Spanish acronym for the Prevention of Childhood Obesity: A Community-Based Model). The entire POIBC protocol has been published elsewhere [12]. In brief, the POIBC intervention study determined the efficacy of the THAO-Child Health Program implemented to prevent childhood obesity in 2249 children aged 8 to 10 years. The POIBC study was an intervention study, with the objective of determining the efficacy of the THAO-Child Health Program [13]. The study was carried out during two academic years (2012/2014), with a mean follow-up of 15 months. After excluding participants with missing data on any of the included variables, a final sample of 1400 children (708 girls and 692 boys) with a mean age of 10.1 ± 0.6 years old was included. All variables were collected at the baseline and follow-up. The local ethics committee (CEIC-PSMAR, Barcelona, Spain; approval number (2011/4296/I)) approved the study. The children were informed about the study and that their participation was voluntary. Written consent was obtained on their behalf from their parents.

### 2.2. Outcome Measures

The anthropometric measurements were assessed by trained personnel on the first day of the intervention at each school. For each participant, the body weight, height, and waist circumference were measured in underwear without shoes using an electronic scale (SECA 813) to the nearest 100 g, a portable stadiometer (SECA 213), and a measuring tape (SECA 201), both to the nearest 1 mm. The age- and sex-specific BMI was calculated as the zBMI, and being overweight and obesity were defined according to IOTF cutoffs [14]. The waist-to-height ratio (WHtR, cm/cm) was calculated; abdominal obesity was defined as a WHtR ≥ 0.50 [15].

### 2.3. Exposure Measures

The children were asked about whether they usually had the following meals: breakfast, a mid-morning snack, lunch, an afternoon snack, and dinner. The response categories were dichotomous (yes/no). For the analyses, the data were categorized into five meals a day, four meals a day, and less than 4 meals a day.

### 2.4. Covariates

Physical activity was measured by the Physical Activity Questionnaire for Children (PAQ-C). The PAQ-C asks about different activities to define the PA level of the previous week (the previous 7 days) [6]. It provides a summary PA score derived from nine items. Each question is scored on a 5-point scale, with higher scores indicating higher levels of activity.

The adherence of children to the Mediterranean diet (MD) was examined using the KIDMED index, which includes 16 items [16]. The range of scores was between −4 and 12, with higher scores reflecting a greater adherence to the MD. The PAQ-C and KIDMED questionnaires were administered in schools with the assistance of trained field researchers at the baseline and follow-up.

The family sociodemographic and lifestyle variables were recorded by parents with corresponding questionnaires. The maternal education level was collected as an indicator of the socioeconomic status and categorized into two levels: (i) university degree; and (ii) less than university degree.

### 2.5. Statistical Analysis

The participant characteristics were described as the mean (standard deviation) and proportions as appropriate, according to the meal frequency categories.

General linear models (GLM) were fitted to analyze the prospective association between the meal frequency categories with the zBMI and the waist-to-height ratio (WHtR). A polynomial contrast was used to estimate the *p* for a linear trend, with a post hoc Bonferroni correction used for the multiple comparisons. The final model was adjusted for sex, age, school, intervention group, maternal education, physical activity, and the corresponding baseline outcome.

The association between the meal frequency and the incidence of excessive weight and abdominal obesity was performed by a logistic regression analysis. For this analysis, we excluded the children with the outcomes at the baseline. The final model was adjusted for the same covariables included in the GLM analysis.

The interactions between the meal frequency and the sex and age were tested.

The associations were considered to be significant if *p* < 0.05. All statistical analyses, with the exception of the dose–response analysis, were performed using SPSS for Windows, version 22 (SPSS, Inc., Chicago, IL, USA).

## 3. Results

No significant interactions between the meal frequency and sex (*p* = 0.665) or age (*p* = 0.872) were found. Most children had 5 meals per day (57.8%), followed by 4 meals (30.1%) and 3 or fewer meals (12.1%) per day.

The baseline characteristics of the study population are shown in Table 1. Boys had a lower meal frequency than girls. Adherence to the MD significantly increased with a higher meal frequency whereas the opposite was found for the proportion of participants with excessive weight. A non-significant borderline trend (*p* = 0.074) was found for a decrease in the zBMI with a higher meal frequency.

Multiple linear regression models adjusted for sex, age, allocation to the intervention group, school, maternal education, physical activity, and for the corresponding outcome variable at the baseline revealed a significant inverse association between the meal frequency with the zBMI and WHtR (Table 2). A further adjustment for adherence to the MD did not affect either the magnitude or the direction of these associations.

Table 3 shows the relationship between the meal frequency and the incidence of excessive weight and abdominal obesity. The odds of developing excessive weight or abdominal obesity during the follow-up independent of the sex, age, allocation to the intervention group, school, maternal education, physical activity, adherence to the MD, and for the corresponding outcome variable at the baseline significantly decreased with an increase in the meal frequency (*p* = 0.035 for excessive weight and *p* = 0.028 for abdominal obesity).

## 4. Discussion

The main finding of the present study was that a higher meal frequency at the baseline was predictive of a lower zBMI and WHtR at the follow-up. Additionally, the incidence of excessive weight and abdominal obesity decreased with an increase in the meal frequency.

In the literature, there is little evidence from intervention studies or prospective cohorts [8,9,11,17,18]. In a one-year randomized controlled trial carried out on defective-hearing children, boys and girls aged 11 to 16 years who ate 7 meals per day instead of 3 presented less weight gain and skinfold thickness (arm, abdomen, and back for both 7 or 5 meals compared with 3 meals) [17]. No statistical significance was observed in the age group 6 to 10 years old [17]. Regarding longitudinal studies, a 10-year prospective study in biracial American girls aged 9 at the baseline reported that girls eating 3+ meals over 3 days had a lower BMI z-score after adjusting for the study site, parental education, race, and indicators of energy intake and expenditure [18]. However, from visit 3 to 10, a change in the eating frequency of the adolescents was reported; girls eating 3+ meals on all 3 days halved whereas girls eating 3+ meals on none of the 3 days almost doubled, weakening the association in the later visits [18]. As adolescents may skip meals as a strategy to control their weight, Ritchie and colleagues repeated the analysis taking into account additional weight-loss diets and found that lower food frequencies remained associated with a higher BMI and waist circumference [11]. In younger children from New Zealand, the meal frequency at 2 years of age did not predict a change in the BMI z-score at 3.5 years after controlling for the intervention group, maternal education, maternal pre-pregnancy BMI, household income, smoking during pregnancy, and weeks of exclusive breastfeeding [9]. In the aforementioned study, a higher percentage of less educated mothers did not provide food diary data for 2 years [9]. As a low maternal education has been associated with a higher risk of childhood obesity, children who were overweight may have been underrepresented, potentially decreasing the significance of the association [19]. Furthermore, in a 9-year prospective study of Peruvian children aged 4 at the baseline, children eating 4 meals a day had a greater increase in the BMI z-score compared with those eating 5 meals a day, although the clinical relevance of this change was small [8]. Younger children, compared with older children, show a greater self-regulation of appetite and are, therefore, less influenced by the meal frequency; hence, perhaps, the discrepancy between our results and those found in younger children [20]. From the age of 10, children enter the preadolescent period and may be more sensitive to external cues such as body image or social desirability/expectations, potentially leading them to engage in unhealthy eating practices such as skipping meals or restricting themselves [21]. Promoting an adequate meal frequency of at least 3 meals (particularly from the age of 10), combined with a good-quality diet and adequate energy intake and expenditure, may be beneficial for the weight status of children, but more studies are needed before drawing early conclusions. Interestingly, most of the aforementioned studies examining the relationship between the meal frequency and weight status did not control for the overall diet quality and energy intake, both of which are related to the development of weight [22,23]. One might expect that higher meal frequencies are related to an increased intake of energy, which potentially increases the probability of an excessive weight gain.

A novel finding in the present study was the inverse linear trend between the meal frequency and the incidence of excessive weight and abdominal obesity. However, the low incidence rate of both outcomes calls for caution in interpreting these findings. There is a clear need for further studies that are adequately powered to answer the question of whether an increased meal frequency is associated with a lower incidence of obesity.

There was little or no change in the association between the meal frequency and the weight outcomes after an additional adjustment for the overall diet quality, which was measured by the adherence to the MD. In a multi-cross-sectional study conducted in American children and adolescents aged 6–11 and 12–19, the meal frequency was positively associated with the Healthy Eating Index (HEI); however, no adjustment was made for the energy intake [24]. Similar results were reported in a study from the UK, but it had a small sample size and did not control for the energy intake either [25]. When adjusting for the latter, a UK study found no significant association between the meal frequency and the MD score. As the MD is not the traditional diet in the UK, it may have been inaccurate to use this questionnaire [12]. The results on the association between the meal frequency and the diet quality are controversial; hence, perhaps, the small changes observed [12,24,25]. Mealtimes are more likely to be enjoyed in a shared family setting where parents may consistently have more control over the quality of the diet of the child [13,26,27]. On the contrary, snack times may be less supervised; therefore, children may be more likely to consume unhealthy food without parents necessarily realizing it [16]. In our study, a mid-afternoon snack was included in the definition of a mealtime. Further studies should assess the differential association between meal and snack frequencies with the BMI z-score of a child whilst ensuring an adequate definition of the word “snacking” [28].

The main limitations of the present study were the lack of an adjustment for the energy intake and the limited statistical power for the analysis of the incidence of excessive weight and abdominal obesity. Additional limitations due to the self-reported data include social desirability and memory bias. The strengths were the relatively large sample size and the direct measurement of the anthropometric variables. Furthermore, this is the first study to assess the association between meal frequencies and incidence of excessive weight and abdominal obesity.

## 5. Conclusions

In conclusion, the findings of the present study showed that higher meal frequencies were predictive for a favorable development of the weight status in Spanish children. Additionally, there was an inverse trend between the meal frequency and the incidence of excessive weight and abdominal obesity. More prospective studies are needed to confirm these findings.

## Figures and Tables

**Table 1 nutrients-15-00870-t001:** Baseline characteristics of the study population across the levels of meal frequency (N = 1400) ^a^.

		Meal Frequency		
	3 or Fewer Meals/d n = 169	4 Meals/d n = 422	5 Meals/d n = 809	*p*-Value for Linear Trend
Sex, boys (%)	103 (60.9)	216 (51.2)	373 (46.1)	<0.001
Age (years)	10.1 ± 0.6	10.1 ± 0.6	10.1 ± 0.6	0.640
Maternal education ^b^	51 (30.2)	156 (37.0)	279 (34.5)	0.581
zBMI	0.84 ± 1.22	0.68 ± 1.13	0.67 ± 1.15	0.074
Excessive weight ^c^	68 (39.1)	129 (30.6)	235 (29.0)	0.022
WHtR (cm/cm)	0.478 ± 0.053	0.472 ± 0.050	0.474 ± 0.054	0.303
Abdominal obesity	60 (35.5)	112 (26.5)	241 (29.8)	0.514
PAQ-C score (unit) ^d^	3.0 ± 0.8	2.9 ± 0.7	3.0 ± 0.7	0.634
KIDMED index (unit) ^e^	6.1 ± 2.5	6.8 ± 2.4	7.1 ± 2.3	< 0.001

^a^ General linear models were fitted to analyze the cross-sectional association between the basal variables and the meal frequency. Values are presented as the number (proportion) and mean (standard deviation) for the categorical and continuously distributed variables, respectively. ^b^ The educational levels of the mother were defined as less than university/more than university. ^c^ Being overweight and obesity, according to the IOTF. ^d^ PAQ-C score ranges from 1 to 5, with a higher score indicating a higher PA level. ^e^ KIDMED index ranges from −4 to 12, with a higher score indicating a higher adherence to the MD.

**Table 2 nutrients-15-00870-t002:** Association between the baseline daily meal frequency with the zBMI and waist-to-height ratio at follow-up ^a^.

	Baseline Meal Frequency			
	3 or Fewer Meals/d	4 Meals/d	5 Meals/d	*p*-Value Linear Trend
At follow-up:	n = 169	n = 422	n = 809	
zBMI				
- Model 1	0.80 (0.62 to 0.97)	0.60 (0.49 to 0.71)	0.57 (0.49 to 0.65)	0.021
- Model 2	0.68 (0.60 to 0.74)	0.62 (0.58 to 0.66)	0.59 (0.56 to 0.61)	0.024
- Model 3	0.78 (0.61 to 0.95)	0.67 (0.61 to 0.73)	0.62 (0.58 to 0.66)	0.016
Waist-to-height ratio (cm/cm)				
- Model 1	0.475 (0.467 to 0.484)	0.463 (0.458 to 0.468)	0.463 (0.459 to 0.467)	0.006
- Model 2	0.471 (0.467 to 0.475)	0.465 (0.463 to 0.467)	0.463 (0.461 to 0.465)	< 0.001
- Model 3	0.471 (0.467 to 0.475)	0.465 (0.463 to 0.468)	0.463 (0.461 to 0.465)	< 0.001

^a^ General linear models were fitted to analyze the prospective association between the basal meal frequencies with the zBMI and the waist-to-height ratio at follow-up. Values are expressed as means (confidence intervals). Model 1 was adjusted for sex and age. Model 2 was adjusted for the covariables in Model 1 and, additionally, for allocation to the intervention group, school, maternal education, physical activity, and for the corresponding outcome variable at baseline. Model 3 was adjusted for the covariables in Model 2 and, additionally, for adherence to the Mediterranean diet.

**Table 3 nutrients-15-00870-t003:** Association between the meal frequency and the incidence of excessive weight and abdominal obesity ^a^.

	Baseline Meal Frequency ^b^			
	3 or Fewer Meals/d	4 Meals/d	5 Meals/d	*p*-Value for Linear Trend
Excessive weight				
- Model 1	Reference	1.05 (0.43 to 2.54)	0.59 (0.25 to 1.41)	0.082
- Model 2	Reference	1.25 (0.46 to 3.40)	0.50 (0.19 to 1.33)	0.025
- Model 3	Reference	1.33 (0.48 to 3.68)	0.54 (0.20 to 1.45)	0.035
Abdominal obesity				
- Model 1	Reference	0.70 (0.36 to 0.76)	0.30 (0.14 to 0.32)	0.009
- Model 2	Reference	0.92 (0.33 to 2.59)	0.37 (0.13 to 1.07)	0.021
- Model 3	Reference	0.92 (0.32 to 2.59)	0.38 (0.13 to 1.11)	0.028

^a^ Multivariate logistic regression analysis. Values are expressed as odds ratio (confidence interval). Excessive weight: being overweight and obesity according to the IOTF; abdominal obesity: waist-to-height ratio > 0.50. Participants in meal frequency categories after eliminating excessive weight cases at baseline. ^b^ 3 or fewer meals/d (reference) n = 103; 4 meals/d n = 294; 5 meals/d n = 575. Participants in meal frequency categories after eliminating abdominal obesity cases at baseline: 3 or fewer meals/d (reference) = 109; 4 meals/d = 311; 5 meals/d = 568. Model 1 was adjusted for sex and age. Model 2 was adjusted for the covariables in Model 1 and, additionally, for allocation to the intervention group, school, maternal education, physical activity, and for the corresponding outcome variable at baseline. Model 3 was adjusted for the covariables in Model 2 and, additionally, for adherence to the Mediterranean diet.

## Data Availability

Data and materials are available upon request to the corresponding authors.

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
