# Peer review of "Association between Meal Frequency and Weight Status in Spanish Children: A Prospective Cohort Study"

_nutrients, 2023, doi:10.3390/nu15040870_

Round 1
Reviewer 1 Report
This is a welcome study, well-conducted and presented. The limitations are highlighted clearly.
Please clarify when the data on diet, meal frequency and anthropometrics were collected? was it only at baseline for diet and meal frequency? and pre and post intervention for anthropometrics?
If all data were collected at baseline and end of intervention, was there a change in meal frequency overtime? and/or any association between the change of meal frequency overtime and BMI at end time point?
As a positive energy balance is the main determinant of BMI, it is indeed regrettable not to have the data on energy intake.
Did the authors collect data on time spent in sedentary behavior (screen time), which has been shown to be correlated with BMI? The physical activity questionnaire may not have been sensitive enough.
Author Response
Reviewer 1
This is a welcome study, well-conducted and presented. The limitations are highlighted clearly.
Reply: Thank you for this assessment of our work.
Please clarify when the data on diet, meal frequency and anthropometrics were collected? was it only at baseline for diet and meal frequency? and pre and post intervention for anthropometrics?
Reply: We have modified the corresponding text in the method section accordingly: “All variables were collected at baseline and follow-up“ (page: 2; line: 87)
If all data were collected at baseline and end of intervention, was there a change in meal frequency overtime? and/or any association between the change of meal frequency overtime and BMI at end time point?
Reply: The reported meal frequencies were relatively stable between baseline and follow-up. Furthermore, an analysis of the relationship between changes of a categorical variable of 5 categories and changes of the outcomes is impossible with the current sample size due to insufficient statistical power. For these reasons we decided to focus on the prospective impact of meal frequency on zBMI and waist to height ratio.
As a positive energy balance is the main determinant of BMI, it is indeed regrettable not to have the data on energy intake.
Reply: Yes, the reviewer is correct. Therefore, we clearly commented this important limitation in the corresponding section of the manuscript (page: 6; lines: 268-269)
Did the authors collect data on time spent in sedentary behavior (screen time), which has been shown to be correlated with BMI? The physical activity questionnaire may not have been sensitive enough.
Reply: We have data on screen time. However, further adjustment of the models (table 2) with this variable did not meaningfully change the magnitude of the association. For this reason, we did not additionally include this variable in the analysis.
Reviewer 2 Report
Dear Authors,
Thank you for your manuscript. The paper is well-written and the study is well-designed and presents important data with the impact on public health.
My comments are mostly technical/minor.
Please explain the abbreviations in the abstract, when they are used first (zBMI, WHtR).
Next, in the abstract age range of the study participants is provided but mean age (± SD) is missing. Please add.
Also, to calculate waist to hips ratio, hips circumference should be also measured. But the description of hips circumference measurement is missing in the section "Outcome measure". Please revise. In addition, an abbreviation WHtR should be also explained (line 88).
Next, in the Methods section it should be specified whether children with the overweight/obesity at the baseline were excluded from the further analyses?
For me it remains unclear whether the study was organized as interventional or observational? From the description of the study organization it seems that the study was organized as observational longitudinal. But in the Abstract, line 37, it is stated that the outcome effects were adjusted for intervention group. Moreover, in line 72 it is also stated that the POIBC is intervention study. Please clarify, because no intervention is presented in your paper.
Also, in the section "Covariables" (Covariates - ?) it should be explained, whether the questionnaires were provided to the parents as well, for example, to collect information on maternal education (lines 104-105).
The statistical analyses are very-well conducted. The results are clearly described and well-discussed.
Finally, it is recommended by the journal requirements to number subsections as well (e.g., 2.1. Study design, 2.2. Outcome measure, etc.).
Author Response
Thank you for your manuscript. The paper is well-written and the study is well-designed and presents important data with the impact on public health.
Reply: We thank the reviewer for this assessment of our work
Please explain the abbreviations in the abstract, when they are used first (zBMI, WHtR).
Reply: This has been done. Waist to Height ratio (WHtR). Standardized BMI (zBMI) (page: 1; line: 36).
Next, in the abstract age range of the study participants is provided but mean age (± SD) is missing. Please add.
Reply: This has been done. (page: 1; lines: 31-32)
Also, to calculate waist to hips ratio, hips circumference should be also measured. But the description of hips circumference measurement is missing in the section "Outcome measure". Please revise. In addition, an abbreviation WHtR should be also explained (line 88).
Reply: We don’t have data on hips circumference. We measured waist circumference and height and calculated with these data the waist to height ratio (WHtR). We have modified the text as follows: “Waist to height ratio (WHtR, cm/cm) was calculated and abdominal obesity was defined as WHtR 0.50 [15].” (page: 2 lines: 97-98)
Next, in the Methods section it should be specified whether children with the overweight/obesity at the baseline were excluded from the further analyses?
Reply: Indeed, an analysis of the incidence of an outcome implies the elimination of the outcome at baseline. This has been stated in the statistic section of the manuscript: “The association between meal frequency and incidence of excessive weight and abdominal obesity was performed by logistic regression analysis. For this analysis we excluded children with the outcomes at baseline.” (page: 3; lines: 135-137).
For me it remains unclear whether the study was organized as interventional or observational? From the description of the study organization it seems that the study was organized as observational longitudinal. But in the Abstract, line 37, it is stated that the outcome effects were adjusted for intervention group. Moreover, in line 72 it is also stated that the POIBC is intervention study. Please clarify, because no intervention is presented in your paper.
Reply: To address the concerns of the reviewer, the study was originally designed as an intervention study, but in this article the data were used for a prospective secondary data analysis. This is a common practice, see for example the PREDIMED or PREDIMED Plus studies among others. We have slightly modified the text in the method study to better describe this fact: “The current study was a prospective cohort in the framework of the POIBC intervention study (Spanish acronym for Prevention of Childhood Obesity: a Community-Based model). The entire POIBC protocol has been published elsewhere [23]. In brief, the POIBC study was an intervention study with the objective to determine the efficacy of the THAO-Child Health Program to prevent childhood obesity in 2249 children aged 8 to 10 years [24].” (page: 2; lines: 81-83).
Also, in the section "Covariables" (Covariates - ?) it should be explained, whether the questionnaires were provided to the parents as well, for example, to collect information on maternal education (lines 104-105).
Reply: We have added the following text to clarify this issue: “Family sociodemographic and lifestyle variables were recorded by parents with corresponding questionnaires” (page: 3 lines: 121,122).
The statistical analyses are very-well conducted. The results are clearly described and well-discussed.
Reply: Thank you.
Finally, it is recommended by the journal requirements to number subsections as well (e.g., 2.1. Study design, 2.2. Outcome measure, etc.).
Reply: This has been done.